# Personalized Medicine in Parkinson’s Disease: New Options for Advanced Treatments

**DOI:** 10.3390/jpm11070650

**Published:** 2021-07-10

**Authors:** Takayasu Mishima, Shinsuke Fujioka, Takashi Morishita, Tooru Inoue, Yoshio Tsuboi

**Affiliations:** 1Department of Neurology, School of Medicine, Fukuoka University, 7-45-1, Nanakuma, Johnan-ku, Fukuoka 814-0180, Japan; mishima1006@fukuoka-u.ac.jp (T.M.); shinsuke@cis.fukuoka-u.ac.jp (S.F.); 2Department of Neurosurgery, School of Medicine, Fukuoka University, Fukuoka 814-0180, Japan; tmorishita@fukuoka-u.ac.jp (T.M.); toinoue@fukuoka-u.ac.jp (T.I.)

**Keywords:** Parkinson’s disease, deep brain stimulation, levodopa-carbidopa intestinal gel, apomorphine, radiofrequency, focused ultrasound, induced pluripotent stem cells, cell therapy, gene therapy, personalized medicine

## Abstract

Parkinson’s disease (PD) presents varying motor and non-motor features in each patient owing to their different backgrounds, such as age, gender, genetics, and environmental factors. Furthermore, in the advanced stages, troublesome symptoms vary between patients due to motor and non-motor complications. The treatment of PD has made great progress over recent decades and has directly contributed to an improvement in patients’ quality of life, especially through the progression of advanced treatment. Deep brain stimulation, radiofrequency, MR–guided focused ultrasound, gamma knife, levodopa-carbidopa intestinal gel, and apomorphine are now used in the clinical setting for this disease. With multiple treatment options currently available for all stages of PD, we here discuss the most recent options for advanced treatment, including cell therapy in advanced PD, from the perspective of personalized medicine.

## 1. Introduction

Personalized medicine is an emerging field that seeks to tailor the treatment of individual patients based on their clinical characteristics, biomarkers, genetics, and other factors [1,2]. Other factors include specific comorbidities, complications, and patient background. To date, personalized medicine in Parkinson’s disease (PD) has not been fully realized due to barriers such as cost and genetic counseling although personalized medicine is used in PD patients in clinical settings when treatments are tailored based on motor and non-motor features [3,4,5,6,7].

PD is a heterogeneous disorder in which motor and non-motor features of varying types and degrees may appear quite separately in individuals [1,8]. Indeed, the etiology and pathogenesis of PD include a mixture of factors without any diagnostically reliable biomarkers; therefore, the diagnosis of PD is still based on a clinical assessment [9,10]. It is known that the prognosis of PD differs between clinical types, with tremor-dominant types progressing slower than postural instability gait difficulty (PIGD) types [11]. The Parkinson’s Progression Markers Initiative (PPMI) clinical study has revealed more detailed subtypes of PD [12]. The authors classified PD into mild motor-predominant, intermediate, and diffuse malignant types [12]. Several studies have been undertaken to address and detect possible biomarkers, which may predict the progression of individual PD patients [13].

Historically, the first PD treatments involved a surgical approach. In 1952, Narabayashi et al. performed the world’s first pallidotomy for PD patients and described its positive effect [14]. In the early 1960s, L-dopa therapy was initiated, but initially, low doses failed to show efficacy in many PD patients; Cotzias then initiated the use of high-dose therapy, and the modern regimen for L-dopa therapy was established [15]. L-dopa is still the gold standard, and its combination with dopamine agonist, monoamine oxidase type B inhibitor, catechol-O-methyltransferase inhibitor and/or non-dopaminergic medication has been used to treat L-dopa related motor and non-motor complications for many years. However, in the advanced stage, despite adjustments to these medications, it is impossible to manage these complications, and finally surgical intervention is required in some patients. The use of stereotactic neurosurgery declined with the introduction of the drug L-dopa as an effective oral medication; but stereotactic neurosurgery was revived when it was shown to be effective in treating motor complications including wearing-off and dyskinesia [16,17]. Later, deep brain stimulation (DBS) was introduced, and became the gold standard of treatment for advanced PD motor features [18]. Today, various advanced treatments such as DBS, radiofrequency, MR–guided focused ultrasound (MRgFUS), gamma knife, levodopa-carbidopa intestinal gel (LCIG), and apomorphine are available, although the availability of treatments varies depending on country and region. Clinical practice guidelines for early treatment of PD have been published in various countries and are often recommended by experts [19,20,21]. Standard pharmacological and non-pharmacological treatments are required during treatment, and the need for personalized medicine becomes more obvious when aiming to achieve an appropriate symptomatic and disease-modifying treatment with the right dose, right time, and minimum side effects in a specific patient. On the other hand, guidelines for the treatment of advanced PD have not been established, and in particular, the indication criteria and exclusion criteria for device-aided therapy have not been clarified. DBS and LCIG are the most established treatments for advanced stage PD in recent years, apomorphine subcutaneous infusion and MRgFUS have also become available, and efforts to incorporate them into personalized medicine will become important in the future. This review focuses on the advanced treatment of PD including cell therapy and gene therapy. Furthermore, we discuss aspects of personalized medicine that are currently available for the advanced treatment of PD.

## 2. Advanced Treatments

In this review, we use the term “advanced treatments” when refering to DBS, LCIG, apomorphine injection, MRgFUS, and other non-medication approaches.

Although the aim of advanced treatment in PD is to improve motor features, this treatment has also been shown to be effective for certain non-motor features [22]. The timing of the introduction of advanced treatments such as DBS or LCIG varies from patient to patient, but, as suggested by Antonini et al. [23], the presence of off-symptoms for more than 2 h a day, troublesome dyskinesia for more than 1 h a day, and levodopa administration of more than 5 times a day may be indicators for advanced PD. The authors described the indications for advanced treatments in PD patients as follows. Patients with good L-dopa response, good cognition, and <70 years of age were considered as good candidates for DBS, LCIG, and apomorphine subcutaneous infusion. More specifically, patients with troublesome dyskinesia can be treated with DBS or LCIG. Patients with L-dopa-resistant tremor were considered good targets for DBS. Previous authors also propose an indicator of which device–aided therapy is appropriate, based on each patient’s background, motor and non-motor features, and activities of daily living by using the Delphi approach [23]. However, with the emergence of new options, it may be necessary to further refine the criteria for personalized treatment. In addition, we should be mindful of whether these advanced treatments are suitable or unsuitable for individual patients on an evidence basis; this currently remains ambiguous.

Currently, or in the near future, the advanced treatment options for PD motor features include/will include DBS, LCIG, apomorphine, MRgFUS, cell therapy, and gene therapy (Figure 1). For medication-resistant tremor associated with PD, the main treatment options are DBS, MRgFUS, radiofrequency, and gamma knife. The characteristics of each treatment for tremor are shown in Appendix A. Below, we focus on and briefly describe the motor features of PD and outline each relevant advanced treatment. Table 1 briefly shows indication, advantages, disadvantages, and adverse effects for DBS, LCIG, and apomorphine, which are currently established advanced treatments for PD.

### 2.1. Deep Brain Stimulation (DBS)

Today, DBS has become one of the most successful surgical treatments in the advanced stages of PD and has been performed in many patients worldwide. During DBS, electrodes are implanted deep in the brain, a pulse generator is implanted in the chest wall, and an electric current is passed through a connected lead wire to stimulate the targeted deep brain tissue (Figure 2). In addition to the selection of the DBS target and the stimulation parameters, new technologies have enabled a personalized approach to PD.

Regarding the brain targets, the subthalamic nucleus (STN) and globus pallidus internus (GPi) are commonly used as targets for DBS in PD. Both targets have their own strengths, and previous studies have compared the therapeutic effects of DBS on motor and non-motor features in both targets. However, as yet, there are no clear criteria for the choice of DBS target for PD patients and this is often determined by the physician’s preference. Negida et al. reviewed the selection between STN and GPi [24]. They report that STN-DBS is preferable from a cost point-of-view, as it allows a greater reduction in anti-Parkinson medication and less battery consumption, while GPi-DBS is better for patients who have problems with mood, speech, or cognition [24].

Other targets are the ventralis intermedius (Vim) and pedunculopontine nucleus (PPN) [25]. Vim-DBS is less effective for bradykinesia and rigidity, but very effective for tremor, and is therefore indicated for PD patients with tremor predominance and minimal motor features other than tremor. Meanwhile, PPN-DBS is effective for postural instability and gait disturbance, and has been suggested to reduce the incidence of falls; however, reported effects are variable [25,26]. In Appendix A, we show the effects of DBS on individual symptoms for each target (STN, GPi, Vim, and PPN). Although there are currently only a few reports, the effects of targeting the post-subthalamic area, or caudal zona incerta (PSA/cZi) are also expected to be positive [27]. Motor features of PD are bilateral in most cases and often have a right/left side dominance. The effectiveness of unilateral STN and GPi-DBS has also been reported [28,29], indicating that unilateral DBS may be an option, especially in cases with a strong left/right dominance. Furthermore, stepped GPi and STN-DBS, which is initially unilateral and then contralateral, or combined unilateral STN and contralateral GPi DBS may offer an effective resolution for certain PD patients [30,31]. It is also noteworthy that the connectomic approach has addressed the identification of stimulation targets in individual cases [32,33], and this technological advancement may also contribute to personalized DBS.

In recent years, with the advancement of DBS technology, directional leads [34,35,36,37,38,39,40] and adaptive DBS (aDBS) [41] have been developed and made clinically available. There are many reports showing the usefulness of directional leads not only in PD but also in essential tremor (ET) [34,35,36,37,38,39,40]. Directional leads can be particularly useful in optimizing STN-DBS stimulation to expand therapeutic windows and avoid stimulation-induced side effects [34]. Krüger et al. showed that tremor was significantly improved after exchange from standard to directional DBS in ET patients. This is the first publication to date that showed a clinical superiority of directional DBS. Thus, directional DBS may have high potential for patients with advanced symptoms [40]. aDBS is a technique that was developed to enable analysis of local field potentials from leads in STN and/or GPi, revealing that beta oscillations are associated with motor features of PD [41]. Conventional DBS conveys sustained stimulation under conditions of constant stimulus, although a change in stimulus is possible. In contrast, aDBS, which uses beta oscillations as an index for control, may have higher therapeutic effects and lower battery consumption than conventional DBS [42]. Research in regulating the stimulation of DBS has also progressed, for example, low-frequency stimulation has been reported to have beneficial effects in patients with “freezing of gait” (FOG) [27]. In addition, recent studies have shown the efficacy of variable stimulation patterns for FOG [43] and cycling mode stimulation for tremor refractory to conventional continuous stimulation patterns [44]. With these new techniques and stimulus adjustments, further improvement of motor and non-motor features in PD patients is expected. Therefore, it is important for clinicians to understand the advantages of devices made by different manufacturers.

Thus, DBS may be the advanced treatment that is most suited to personalized medicine. Clinical teams should be aware that selection of the optimal brain target(s), device, and the stimulation parameters are all critically important. It is necessary to decide the optimal indication for surgical treatment according to the timing of treatment and an individual’s unmet needs. In addition, most patients on whom surgery is performed are in an advanced stage of PD; therefore, support such as medical management, exercise therapy, and a suitable living environment are required even after DBS treatment. Motor complications are also indications for DBS. The advantage is that it does not require a caregiver, as shown in Table 1 above; however, the disadvantage is the possibility of psychiatric and cognitive changes. Multidisciplinary team medical care is a major driver behind solving these problems. This will be described in detail later.

We discuss potential treatments at the end of this section. Optogenetics is technology to control the functions of neurons by using genetically coded, light-gated ion channels or pumps, and light. This biological technique has contributed to our understanding of nervous system function. Although the application of optogenetics to non-human primates is limited, Watanabe et al. shows that neural activity and behavior in non-human primates can be manipulated optogenetically [45]. These studies may also lead to applications for DBS. In addition, the evolving technologies of magnetogenetics, which manipulating neurons with magnetic stimuli, and sonogenetics, which focuses on the genetic modulation of ultrasound-sensitive neurons and their specific responses to ultrasound, could contribute to the advanced treatment of PD for the possibility of being minimally invasive [46,47].

### 2.2. Levodopa-Carbidopa Intestinal Gel (LCIG)

Continuous dopaminergic delivery is required to resolve motor complications that are problematic in advanced PD patients. In addressing this situation, the mechanism of LCIG is ideal: it involves continuous infusion of levodopa directly into the jejunum (Figure 3), where it is absorbed via a transgastrostomal jejunal tube that maintains a constant blood levodopa concentration, thereby reducing motor complications [48]. The effect on motor complications such as reduction in off-time per day can be maintained for a lengthy period [49]. It is also effective in the treatment of cases of FOG that are resistant to pharmacological treatment [50]. LCIG is reported to improve non-motor features such as anxiety, sleep disorders, depression, hallucinations, impulse control disorders, and cognition [49,51,52]; however, there is less evidence than for its effects on motor features, so more research is needed in the future. The frequency of complications with LCIG is high [53]. Surgery-related complications include pain, gastrointestinal symptoms, and device failure, most of which decrease in frequency by two weeks post-surgery [53]. In addition to device failure, weight loss, cholecystitis, and neuropathy are complications of the long-term course [54,55,56]. It is necessary to check each patient’s background before introducing an LCIG device, as, if the patient has difficulty with its use, a caregiver may be needed. The optimal indication for LCIG also needs to be determined. A multi-disciplinary medical team can be very helpful in advancing this treatment.

### 2.3. Apomorphine

Apomorphine, a dopamine agonist, is administered through subcutaneous rescue injection or subcutaneous infusion. Rescue injection is an established rescue therapy for patients with PD associated with motor fluctuations [57,58]. Katzenschlager et al. describes the efficacy of apomorphine subcutaneous infusion in patients with PD with motor fluctuations through the presentation of a multicenter, double-blind, randomized, placebo-controlled trial in 2018 [59]. This has now become one of the advanced treatment options for PD, along with DBS and LCIG. The indications for apomorphine are motor complications; this is a minimally invasive procedure compared to DBS and LCIG, as shown in Table 1 above; however, if the patient is unable to operate the device, a caregiver may be required. In Japan, rescue injection is available, but subcutaneous infusion is not, so further expansion of the treatment is expected in the near future.

### 2.4. Ablative Surgery

#### 2.4.1. Radiofrequency Lesioning

Radiofrequency is the oldest surgical treatment for PD and was a cornerstone of the development of DBS. Radiofrequency thalamotomy is an established treatment for tremor. Tasker compares the efficacy and complications of radiofrequency thalamotomy and DBS for symptoms of tremor [60]. This study shows that DBS is more costly and requires more management, but DBS has fewer complications than radiofrequency thalamotomy because of the need to adjust stimulation parameters in DBS [60]. More recently, DBS has become the preferred choice over radiofrequency for tremor. Complications of both radiofrequency and DBS include cerebral hemorrhage. Radiofrequency thalamotomy can be repeated in cases of tremor recurrence, and additional DBS may be an option [60]. Schreglmann et al. reviews functional neurosurgery for tremor [61]. The authors indicate that when comparing the size of lesions following treatment with radiofrequency or MRgFUS, at 12 months after surgery, the size of lesions undergoing radiofrequency may be greater than that of FUS [61]. A study examining the recurrence rate of MRgFUS in patients with essential tremor shows that the recurrence rate decreases with increased lesion size [62]. Thus, at this time, radiofrequency may be less likely to result in recurrence than MRgFUS. For PD patients who are against the use of an implanted device for cosmetic reasons, thalamotomy is an alternative treatment option for tremor.

#### 2.4.2. Gamma Knife

Similar to MRgFUS, gamma knife does not require burr hole craniotomy and is considered as a minimally invasive treatment; however, it does not allow the intraoperative observation of symptoms. In addition, physicians should be cautious that this therapy may result in late cyst formation and/or radiation necrosis in some cases as a high level of radiation is required for the treatment. Unilateral gamma knife thalamotomy has been shown to be effective in treating tremor in PD [63]. In addition, studies on the motor features of PD following the use of gamma knife pallidotomy and subthalamic gamma knife radiosurgery have been investigated [64,65]. Unilateral gamma knife thalamotomy is a potential alternative to DBS and radiofrequency thalamotomy for tremor in PD patients with contraindications for surgery [63]; however, due to the success and increased use of MRgFUS, the latter treatment may replace gamma knife in the future when MRgFUS overcomes the current technical issues because of the possibility of secondary neoplasia due to radiation exposure and difficulty in detecting complications during the procedure, due to the time for the treatment to take effect.

#### 2.4.3. MR–Guided Focused Ultrasound (MRgFUS)

MRgFUS is a treatment that has recently received tremendous attention. FUS was originally difficult to apply for intracranial diseases due to the attenuation and scattering of ultrasound in the skull, but advances in technology have overcome these problems. MRgFUS can be repositioned, or treatment discontinued depending on the neurological condition of the patient being treated. It does require the total shaving of the patient’s head, but it does not require the burr hole craniotomy that is needed for DBS or radiofrequency. Thus, MRgFUS is considered a minimally invasive therapy (Appendix A). However, physicians should be cautious that the incidence of permanent complications of MRgFUS may be higher than DBS due to the nature of lesioning [66]. For example, a recent randomized trial of MRgFUS subthalamotomy reveals a complication rate as high as 25%, including gait and speech disturbance as well as new onset of dyskinesia [67]. The complications reported in the same study are consistent with conventional radiofrequency subthalamotomy, despite the fact that subthalamotomy is performed unilaterally [68], so clinicians should be aware that any form of subthalamotomy may result in similar problems.

Bond et al. report the suppression of tremor following the application of unilateral MRgFUS thalamotomy in patients with PD [69]. Regarding other targets, MRgFUS subthalamotomy and pallidothalamic tractotomy for PD lead to the improvement of MDS UPDRS or UPDRS Part Three scores [67,70]. Based on these studies it is hoped that, in the not too distant future, this treatment will have an effect not only on tremor but also on other motor features. Furthermore, research into the relationship between lesion size and clinical outcome will help establish more optimal treatment methods. Because of concerns regarding complications of bilateral treatment of MRgFUS, it is a good indication for patients with prominent unilateral symptoms or tremor, and it is therefore thought to have the advantage over DBS therapy at this stage in patients who need improvement in unilateral symptoms [66,71].

### 2.5. Comparison of DBS and LCIG

A meta-analysis was performed based on comparisons between STN-DBS and LCIG [72]. In this study, no significant differences were noted between STN-DBS and LCIG on UPDRS Part Three and adverse events [72]. Furthermore, the results show no significant difference in motor features in the overall therapeutic effect of each surgical treatment. Moreover, EUROPAR and the International Parkinson and Movement Disorders Society Non-Motor Parkinson’s Disease Study Group examined motor and non-motor features in STN-DBS, LCIG, and apomorphine [73]. The latter study, based on an eight-item Parkinson’s disease questionnaire (PDQ-8), UPDRS Part Four, and NMSScale, reveals that total scores were improved significantly in all groups. The authors highlight the importance of holistic assessments to personalize treatment choices [73]. We show the advantages and disadvantages of DBS and LCIG from a perspective of holistic assessments in Appendix A.

### 2.6. Combination Therapy

In their study, Elkouzi et al. report a case series of advanced PD patients treated with DBS and LCIG [74]. Six patients were treated with DBS (bilateral STN DBS, bilateral GPi DBS, and unilateral GPi DBS) who subsequently received rescue LCIG therapy. Following this treatment, an improvement in the 39-item Parkinson’s disease questionnaire (PDQ-39) was noted for four patients. The authors went on to propose an algorithm for the potential use of rescue LCIG therapy in PD-DBS patients. Therefore, PD-DBS patients with persistent or recurrent motor fluctuations who have difficulty with further DBS interventions may be candidates for additional LCIG treatment [74]. In addition to dual DBS and LCIG therapy, other surgical treatment combinations may be useful in selected cases, but cost does need to be considered.

### 2.7. Future Surgical Treatments

#### 2.7.1. Cell Therapy

Since the 1980s, fetal dopaminergic transplantation has been performed in patients with PD and studies report an improvement in motor features following this treatment [75,76]. However, fetal dopaminergic transplantation encountered problems with ethical issues, including difficulty in obtaining sufficient amounts of fetal brain tissue, and contamination of serotonin neurons with associated dyskinesia. These problems have been solved following the introduction of induced pluripotent stem cell (iPSC) technology. Indeed, a primate study shows significant improvement two years after transplantation of human iPSCs into a primate PD model [77]. Human transplantation into PD patients was first practiced in Japan [78], where allogeneic transplantation is now performed [78]. In contrast, Schweitzer et al. performed autologous transplantations [79]; they report no significant change in MDS-UPDRS Part Three scores; however, they noted an improvement in PDQ-39 [79]. In autologous transplantation, if the patient has genetic variants, iPSCs are genome edited and differentiated into midbrain dopaminergic progenitor cells, which can then be transplanted (Figure 4). On the other hand, allogeneic transplantation requires immunosuppressive drugs; it is also advisable to check that the donated cells do not have genetic variants. Figure 4 shows the process of cell therapy in patients with PD using iPSCs. Drug treatment and rehabilitation are still needed in cases of cell therapy [80], and the collection of data from a greater series of cases is necessary to truly reflect the effectiveness of cell therapy using iPSCs.

#### 2.7.2. Gene Therapy

Dopamine deficiency in the putamen causes motor features in PD. Therefore, gene therapy has been adopted to replenish dopamine by introducing genes of enzymes necessary for dopamine synthesis into neurons in the putamen [81]. An adeno-associated virus (AAV) vector has been the most commonly used gene therapy for PD patients in clinical trials, although an equine infectious anemia virus (EIAV) has also been used [81,82,83,84,85,86]. Muramatsu et al. [81] and Christine et al. [82] report that AAV vectors expressing aromatic-amino acid decarboxylase (AADC) were administered to the putamen of PD patients, and the patients subsequently showed improved UPDRS Part Three scores. Christine et al. further administered higher doses of AAV vectors to PD patients and showed increasing on-time in PD patients [83]. Gene therapy, implemented by injecting EIAV vectors carrying the three genes (tyrosine hydroxylase, AADC, and GTP-cyclohydrolase 1) into the putamen of PD patients, has also been performed [84]. Furthermore, gene therapy employing transfer of the trophic factor neurturin into the putamen [85] and glutamic acid decarboxylase into the STN [86] via an AAV vector have been conducted. The huge benefit of gene therapy is that it does not require immunosuppressive drugs, which are necessary for allogeneic cell transplantation using iPSCs; in addition, the mass production of vectors is possible. Further research is needed to determine targets, dose, and which genes to introduce for the practical application of treatment in PD patients.

## 3. Evaluation of the Efficacy of Advanced Treatment

Since there are no disease-modifying treatments for PD, the current goal of PD treatment is to improve patient and caregiver satisfaction. Physicians may tend to focus on the improvement rate of MDS UPDRS Part Three scores when evaluating the effectiveness of advanced treatment. However, the possibility of a gap between physician evaluation of surgical treatment effectiveness and patient and caregiver satisfaction should be noted; despite this, few studies have examined patient satisfaction with advanced treatment for PD [87]. A large multicenter study of PD patients showed that MDS UPDRS Parts One and Two affect their quality of life (QOL) [88]. Although the short-form PDQ-8 and the PDQ-39 have been used in many studies [88,89,90], MDS UPDRS Parts One and Two, the patient reported outcome (PRO)-based assessments of patients’ activities of daily living (ADL), is also useful in the assessment of advanced treatment. Regarding non-motor features, the Non-Motor Symptoms Scale for Parkinson’s Disease (NMSS), the Non-Motor Symptoms Questionnaire (NMSQ), and MDS Non-Motor Rating Scale (MDS-NMS) may be useful for evaluating end points of advanced treatment. Furthermore, it is expected that outcomes assessed by caregivers [91] will also be used to judge the effectiveness of advanced treatment of PD.

## 4. Team Approach

Organization of multidisciplinary clinical care teams is recommended in PD treatment [92], and a team approach is essential for the realization of personalized medicine for advanced treatment in PD patients. An example of a team approach to advanced treatment of PD, particularly LCIG and stereotactic neurosurgery, is presented in Figure 5. Neurologists take a lead in determining treatment plans, but neurosurgeons are responsible for stereotactic neurosurgery, and gastroenterologists and colorectal surgeons are responsible for LCIG. Furthermore, psychiatrists are important in the evaluation and treatment of psychiatric symptoms, and dentists are needed to evaluate and care for dysphagia which is frequently seen in PD. Therapists play an important role in sustained rehabilitation, and assessment of ADL requires cooperation with therapists. The presence of a nurse is important for assessment of the patient’s background, and PD nurses [92] are indispensable during the long process of advanced treatment. Caregivers as well as patients require nursing care. Pharmacist medication guidance is also important for the continuous treatment of various drugs. Higuchi et al. reveals that screening through the use of a team approach may be useful for more than just patient selection of DBS [93]. Appendix A (DBS) and Appendix A (LCIG) show concerns from a multidisciplinary perspective in determining indications for advanced treatment of PD patients. Since any advanced treatment is invasive, patients may expect notable effects of such treatment in return, which may lead to reduced patient satisfaction [94]. Multidisciplinary informed consent is needed from patients and caregivers when advanced treatment is indicated. The above-mentioned improvement in QOL following cell therapy using iPSCs [79] may also benefit patient satisfaction with a team approach. Moreover, a team approach will be increasingly necessary in the implementation of cell therapy and gene therapy, which are expected to become more widespread in the near future.

A team approach also enables a tailored treatment plan for each patient based on patient-specific risks versus benefit analyses, accessibility to the center, supportive care circumstances, and cultural background. For example, surgical procedures requiring general anesthesia are contraindicated in patients with severe cardiopulmonary risks. Living in a remote area or poor supportive care circumstances may jeopardize LCIG, which requires daily medication renewal. Concerning cultural background, some patients may have a stigma against the use of devices, and in such cases lesion therapy and/or cell therapy may be a suitable option. Additionally, select patients may benefit from a combination of multiple treatment modalities (e.g., unilateral DBS and contralateral RF lesioning). We consider that a team approach at an experienced center would maximize the benefit of tailor-made treatment effects in the application of surgical procedures.

COVID-19 has led to major changes in medical systems globally [95]. It affects PD patients and particularly those that may have lost healthcare resources during the period of the pandemic [96]. During this period, the use of telemedicine, which is recommended in PD treatment, has been useful for outpatient care and may be continued into the future [97]. We conducted a questionnaire survey regarding telemedicine among PD patients in Japan [98]. The results revealed that a majority of patients were aware of the availability of this means of healthcare. Smartphone users, credit card users, and those who lived in regions distant from a hospital tend to prefer the convenience of this facility [98]. Although individual situations vary between countries and regions, telemedicine may be useful for continuing outpatient treatment of PD patients who have undergone advanced treatment. Indeed, the usefulness of telemedicine has been reported in DBS and LCIG even before the COVID-19 pandemic [99,100]. The spread of telemedicine may have a great impact on the choice of advanced treatment for PD.

## 5. Conclusions

Here, we have discussed various advanced treatments for advanced PD. In the future, there may be additional advanced treatment options, including cell therapy and gene therapy. In addition, the development of optogenetics, magnetogenetics, and sonogenetics is expected. Therefore, it is important to consider the individual symptoms, patient background, and cost of these options when deciding on advanced treatment.

## Figures and Tables

**Figure 1 jpm-11-00650-f001:**
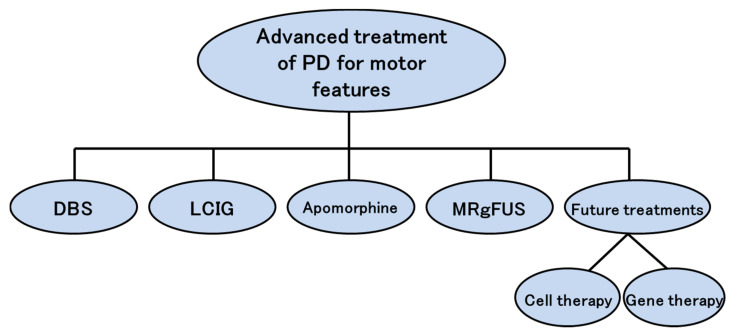
Advanced treatment for motor features of Parkinson’s disease. PD: Parkinson’s disease; DBS: deep brain stimulation; LCIG: Levodopa-carbidopa intestinal gel; MRgFUS: MR–guided focused ultrasound.

**Figure 2 jpm-11-00650-f002:**
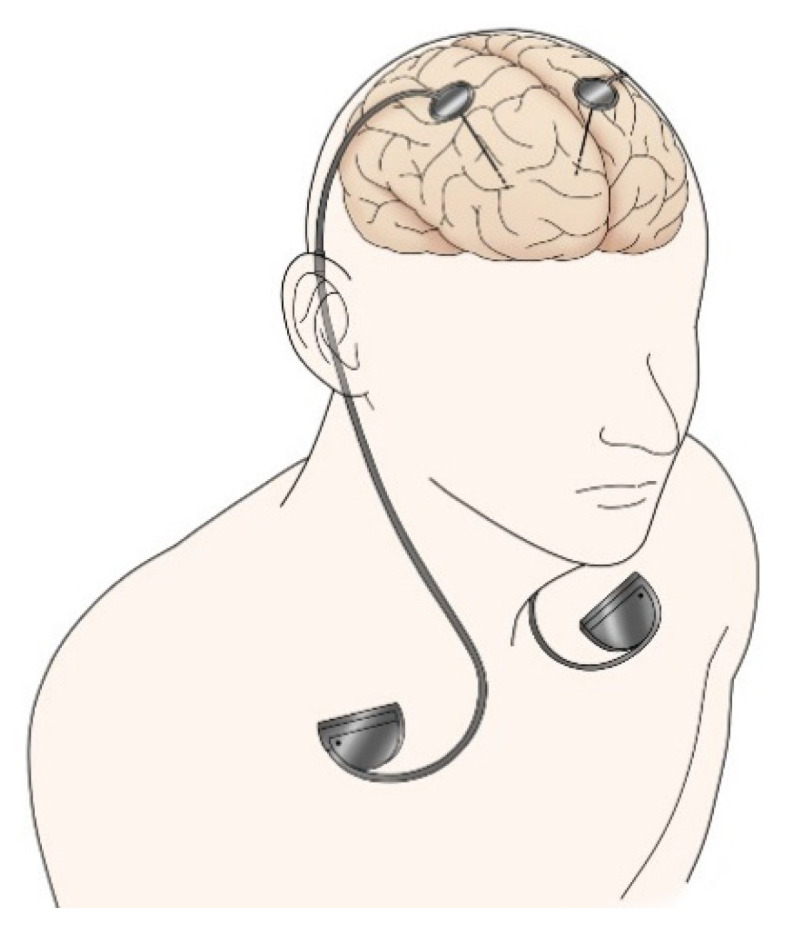
Deep brain stimulation (DBS).

**Figure 3 jpm-11-00650-f003:**
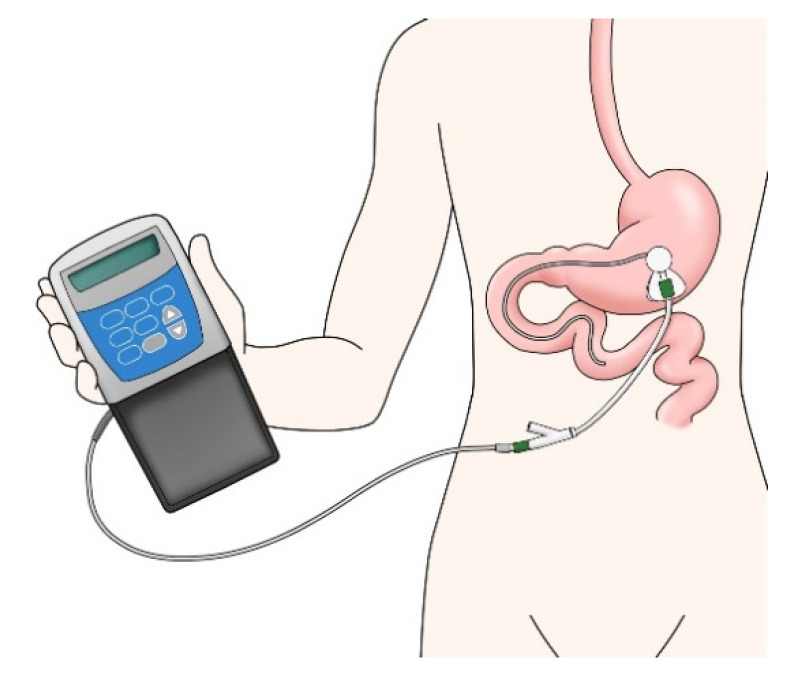
Levodopa-carbidopa intestinal gel (LCIG).

**Figure 4 jpm-11-00650-f004:**
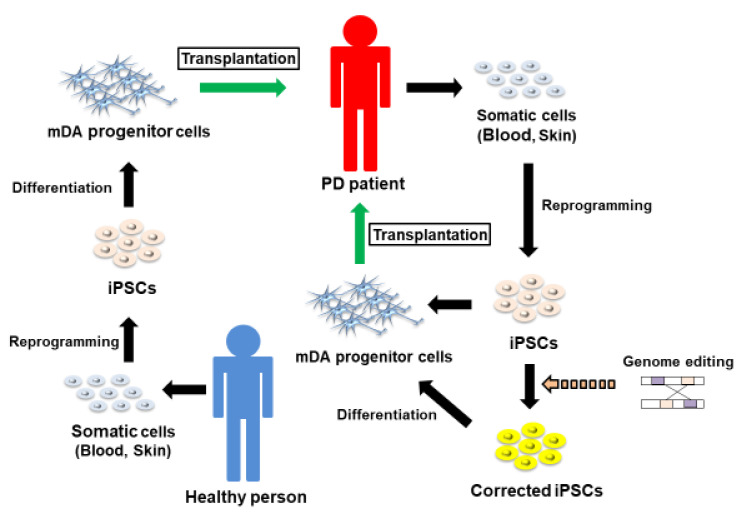
A schema showing cell therapy in patients with Parkinson’s disease using induced pluripotent stem cells (iPSCs). PD: Parkinson’s disease; mDA: midbrain dopaminergic.

**Figure 5 jpm-11-00650-f005:**
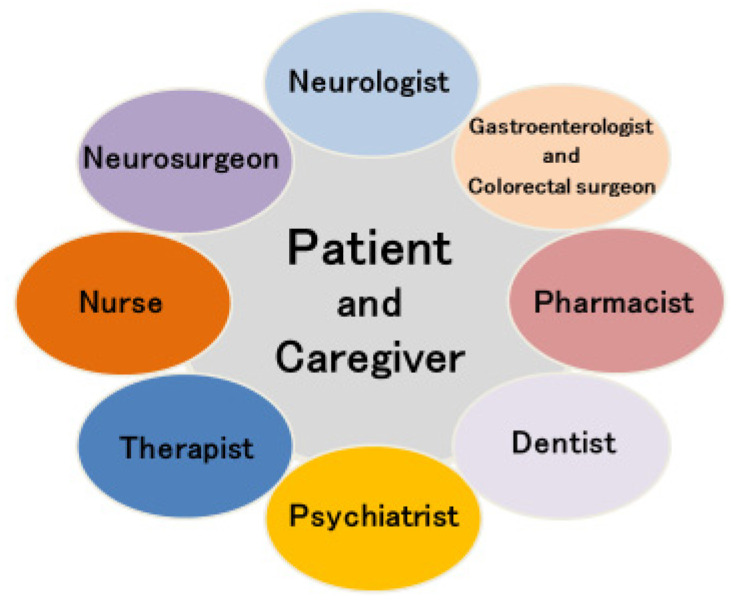
Team approach of advanced treatment for Parkinson’s disease.

**Table 1 jpm-11-00650-t001:** Comparison of different advanced treatments.

	DBS	LCIG	Apomorphine
Indication	Motor complications (especially dyskinesia)	Motor complications	Motor complications (especially motor fluctuations)
Advantages	Dopaminergic medication reduction	No age limit	Minimally invasive procedures
Disadvantages	Invasive procedures	Requires caregivers to handle devices	Requires caregivers to handle devices
Adverse effects	Psychiatric and cognitive changes	Tube trouble	Skin reaction or trouble

DBS: deep brain stimulation; LCIG: Levodopa-carbidopa intestinal gel.

## Data Availability

All relevant data are included in the study and Appendix A.

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
