# Peer review of "Personalized Medicine in Parkinson’s Disease: New Options for Advanced Treatments"

_jpm, 2021, doi:10.3390/jpm11070650_

Round 1

Reviewer 1 Report

This paper has been thoroughly revised and has improved in many aspects. It now has a clear structure and focuses on the most important techniques.

However, there are, a few  aspects that could be improved:

1. The literature on directional electrodes is scarce. The authors mention one paper by Eleopra et al., however, there are numerous more and more important papers that should be mentioned and can be summarized in one additional sentence (see below):

Shao MM, Liss A, Park YL et al. Early experience with new generation deep brain stimulation leads in Parkinson’s disease and essential tremor patients. Neuromodulation Technol Neural Interface. 2020;0:537–542. https://doi.org/101111/ner.13034.

Pollo C, Kaelin-Lang A, Oertel MF et al. Directional deep brain stimulation: an intraoperative double-blind pilot study. Brain 2014;137:2015–2026. https://doi.org/10.1093/brain/awu102.

Contarino MF, Bour LJ, Verhagen R et al. Directional steering: a novel approach to deep brain stimulation. Neurology 2014;83:1163–1169. https://doi.org/10.1212/WNL.0000000000000823.

Steigerwald F, Müller L, Johannes S, Matthies C, Volkmann J. Directional deep brain stimulation of the subthalamic nucleus: a pilot study using a novel neurostimulation device. Mov Disord 2016;31:1240–1243. 

Dembek TA, Reker P, Visser-Vandewalle V et al. Directional DBS increases sideeffect thresholds—a prospective, double-blind trial. Mov Disord 2017;32: 1380–1388. https://doi.org/10.1002/mds.27093.

Also, it would be worth mentioning that for ET patients Krüger et al. (Neuromodulation 2020) showed that tremor was significantly improved after exchange of sDBS for dDBS. This is the first publication to date that showed a clinical superiority of dDBS. Thus, dDBS has high potential for patients with advanced symptoms.

  1. Page 8, lines 290 to 294 are misleading.

What are you trying to say with the last two sentences? Please clarify.First you say MRgFUS is better for patients with unilateral symptoms. Then you say it has an advantage over DBS for patients who need bilateral treatment. This is confusing.

  1. Conclusion:

The last paragraph on optogenetics should not be part of the conclusion since it wasn`t mentioned before. The conclusion should conclude on those aspects discussed before but should not bring up a completely new topic. Also, you would need to explain what magnetogenetics and sonogenetics are. Please integrate it at the end of the main part and then add a sentence on in the conclusion at max.

Author Response

Reviewer 1, comment:

  1. The literature on directional electrodes is scarce. The authors mention one paper by Eleopra et al., however, there are numerous more and more important papers that should be mentioned and can be summarized in one additional sentence (see below):

Shao MM, Liss A, Park YL et al. Early experience with new generation deep brain stimulation leads in Parkinson’s disease and essential tremor patients. Neuromodulation Technol Neural Interface. 2020;0:537–542. https://doi.org/101111/ner.13034.

Pollo C, Kaelin-Lang A, Oertel MF et al. Directional deep brain stimulation: an intraoperative double-blind pilot study. Brain 2014;137:2015–2026. https://doi.org/10.1093/brain/awu102.

Contarino MF, Bour LJ, Verhagen R et al. Directional steering: a novel approach to deep brain stimulation. Neurology 2014;83:1163–1169. https://doi.org/10.1212/WNL.0000000000000823.

Steigerwald F, Müller L, Johannes S, Matthies C, Volkmann J. Directional deep brain stimulation of the subthalamic nucleus: a pilot study using a novel neurostimulation device. Mov Disord 2016;31:1240–1243.

Dembek TA, Reker P, Visser-Vandewalle V et al. Directional DBS increases side effect thresholds—a prospective, double-blind trial. Mov Disord 2017;32: 1380–1388. https://doi.org/10.1002/mds.27093.

Also, it would be worth mentioning that for ET patients Krüger et al. (Neuromodulation 2020) showed that tremor was significantly improved after exchange of sDBS for dDBS. This is the first publication to date that showed a clinical superiority of dDBS. Thus, dDBS has high potential for patients with advanced symptoms.

Response to Reviewer 1:

We thank Reviewer 1 for the insightful comments. We have added the following sentences.

There are many reports showing the usefulness of directional leads not only in PD but al-so in essential tremor (ET) [34-40].

Krüger et al. showed that tremor was significantly improved after exchange from standard to directional DBS in ET patients. This is the first publication to date that showed a clinical superiority of directional DBS. Thus, directional DBS may have high potential for patients with advanced symptoms [40].

Reviewer 1, comment:

Page 8, lines 290 to 294 are misleading.

What are you trying to say with the last two sentences? Please clarify. First you say MRgFUS is better for patients with unilateral symptoms. Then you say it has an advantage over DBS for patients who need bilateral treatment. This is confusing.

Response to Reviewer 1:

We agree with Reviewer 1’s comments. We have revised the sentence as below.

it is therefore thought to have the advantage over DBS therapy at this stage in patients who need improvement in bilateral→unilateral symptoms [57, 62].

Reviewer 1, comment:

Conclusion:

The last paragraph on optogenetics should not be part of the conclusion since it wasn`t mentioned before. The conclusion should conclude on those aspects discussed before but should not bring up a completely new topic. Also, you would need to explain what magnetogenetics and sonogenetics are. Please integrate it at the end of the main part and then add a sentence on in the conclusion at max.

Response to Reviewer 1:

We would like to thank Reviewer 1’s comments. We have moved sentences to the main part (2.1. Deep Brain Stimulation (DBS) section), and added just one sentence to the conclusions.

DBS section

We discuss potential treatments at the end of this section. Optogenetics is technology to control the functions of neurons by using genetically coded, light-gated ion channels or pumps, and light. This biological technique has contributed to our understanding of nervous system function. Although the application of optogenetics to non-human pri-mates is limited, Watanabe et al. shows that neural activity and behavior in non-human primates can be manipulated optogenetically [45]. These studies may also lead to applica-tions for DBS. In addition, the evolving technologies of magnetogenetics, which manipu-lating neurons with magnetic stimuli, and sonogenetics, which focuses on the genetic modulation of ultrasound-sensitive neurons and their specific responses to ultrasound, could contribute to the advanced treatment of PD for the possibility of being minimally invasive [46, 47].

Conclusions

In addition, the development of optogenetics, magnetogenetics, and sonogenetics is expected.

Reviewer 2 Report

The authors have addressed satisfactorily all the issues raised.

I have some further suggestions (minor):

“To date, personalized medicine has not been realized in the clinical setting due to barriers such as cost and genetic counselling.”. I am not sure that, in general, personalized medicine has not been realized (i.e. oncology). To some extent, personalized medicine is also used in PD patients in clinical settings when treatments are tailored based on motor and non-motor symptoms.  I would recommend editing that sentence.

Line 121- Delphi consensus: The authors should mention the '5-2-1' screening criteria (≥five-times daily oral levodopa use, ≥two daily hours with 'Off' symptoms or ≥one daily hour with troublesome dyskinesia) and its applicability to select patients for advanced treatments.

This sentence should be rephrased: “ […] and it is therefore thought to have the advantage over DBS therapy at this stage in patients who need improvement in bilateral symptoms […]. It is not clear what the authors mean.

In general, it is recommendable to use the term “motor features” instead of “motor symptoms”. In fact, motor features are both sign and symptoms.

Section ”Evaluation of the efficacy of surgical advanced treatment”: the authors should expand on the possibility to evaluate as end-points also non-motor symptoms changes using adequate and validated scales: example NMSS, NMQuest and the new MDS-NMS.

Author Response

Reviewer 2, comment:

“To date, personalized medicine has not been realized in the clinical setting due to barriers such as cost and genetic counselling.”. I am not sure that, in general, personalized medicine has not been realized (i.e. oncology). To some extent, personalized medicine is also used in PD patients in clinical settings when treatments are tailored based on motor and non-motor symptoms.  I would recommend editing that sentence.

Response to Reviewer 2:

We thank Reviewer 2 for the insightful comments. We have revised the sentence as you recommend as below.

To date, personalized medicine in Parkinson's disease (PD) has not been fully realized due to barriers such as cost and genetic counseling although personalized medicine is used in PD patients in clinical settings when treatments are tailored based on motor and non-motor features [3-7].

Reviewer 2, comment:

Line 121- Delphi consensus: The authors should mention the '5-2-1' screening criteria (≥five-times daily oral levodopa use, ≥two daily hours with 'Off' symptoms or ≥one daily hour with troublesome dyskinesia) and its applicability to select patients for advanced treatments.

Response to Reviewer 2:

We would like to thank Reviewer 2’s comments. We have added the following sentences.

The authors described the indications for advanced treatments in PD patients as follows. Patients with good L-dopa response, good cognition, and <70 years of age were considered as good candidates for DBS, LCIG, and apomorphine subcutaneous infusion. More specifically, patients with troublesome dyskinesia can be treated with DBS or LCIG. Patients with L-dopa-resistant tremor were considered good targets for DBS.

Reviewer 2, comment:

This sentence should be rephrased: “ […] and it is therefore thought to have the advantage over DBS therapy at this stage in patients who need improvement in bilateral symptoms […]. It is not clear what the authors mean.

Response to Reviewer 2:

We agree with Reviewer 2’s comments. We have revised the sentence as below.

it is therefore thought to have the advantage over DBS therapy at this stage in patients who need improvement in bilateral→unilateral symptoms [57, 62].

Reviewer 2, comment:

In general, it is recommendable to use the term “motor features” instead of “motor symptoms”. In fact, motor features are both sign and symptoms.

Response to Reviewer 2:

We thank Reviewer 2 for the insightful comments. We have changed “motor symptoms” to“motor features”.

Reviewer 2, comment:

Section ”Evaluation of the efficacy of surgical advanced treatment”: the authors should expand on the possibility to evaluate as end-points also non-motor symptoms changes using adequate and validated scales: example NMSS, NMQuest and the new MDS-NMS.

Response to Reviewer 2:

We would like to thank Reviewer 2’s comments. We have added the following sentences as you recommended. “Surgical treatment” has been changed to “advanced treatment” in some parts of the text.

Regarding non-motor features, the Non-Motor Symptoms Scale for Parkinson's Disease (NMSS), the Non-Motor Symptoms Questionnaire (NMSQ), and MDS Non-Motor Rating Scale (MDS-NMS) may be useful for evaluating end points of advanced treatment.

Round 2

Reviewer 1 Report

The manuscript has improved with the changes made by the authors. However, there are some further aspects that need to be improved, that also affect these new changes.

  • Some sentences are in present, some in past sentence. Please make this the same (e.g. p 2 l. 91 authors described, p3 l. 96: previous authors also propose.) Please check the rest of the manuscript to make sure it is in the same tense.
  • P 3, l. 96 has to be good “candidates” not good “targets”.
  • I don`t understand the introduction sentence to optogenetics on page 5, l. 194. There is no further discussion on this lateron. Please exchange for an introductory sentence that might also explain why you chose to put it at the end of the DBS section/ why or that this technology might have an impact on DBS.
  • Please explain how you performed your literature search and decided, which paper to include into this review?